# Circulating IgG Fragments for Gastric Cancer and Esophageal Cancer

**DOI:** 10.3390/diagnostics14131396

**Published:** 2024-06-30

**Authors:** Eugene I. Goufman, Nataliia B. Tikhonova, Andrey P. Aleksankin, Karina B. Gershkovich, Alexander A. Stepanov, Irina I. Stepanova, Liudmila M. Mikhaleva, Natalia V. Nizyaeva, Olga V. Kovaleva, Alexander A. Alferov, Yury B. Kuzmin, Nikolay E. Kushlinskii

**Affiliations:** 1Avtsyn Research Institute of Human Morphology of Federal State Budgetary Scientific Institution “Petrovsky National Research Centre of Surgery”, 117418 Moscow, Russia; morpholhum@list.ru (N.B.T.); aap2004@yandex.ru (A.P.A.); 9163407056@mail.ru (A.A.S.); i-ste@yandex.ru (I.I.S.); mikhalevalm@yandex.ru (L.M.M.); niziaeva@gmail.com (N.V.N.); 2N. M. Emanuel Institute for Biochemical Physics, Russian Academy of Sciences, 119334 Moscow, Russia; gkb-09@mail.ru; 3Federal State Budgetary Institution «N.N. Blokhin National Medical Research Center of Oncology», 115478 Moscow, Russia; ovkovaleva@gmail.com (O.V.K.); aleksandr.alferov@yahoo.com (A.A.A.); yriikuzmin@yandex.com (Y.B.K.); biochimia2023@mail.ru (N.E.K.)

**Keywords:** gastric cancer, esophageal cancer, tumor proteolysis activity, IgG fragments with C-terminal lysine

## Abstract

Blood serum of patients with gastric (*n* = 68) and esophageal (*n* = 43) cancer was assessed for proteolytic fragments of IgG. Serum samples of 20 healthy donors were used as a control. We analyzed indicators of hemostasis (prothrombin time, fibrinogen, plasminogen activity, a2-antiplasmin activity, protein C activity) in blood plasma and the level of total IgG in the blood serum. The median IgG-LysK of healthy donors was lower than in esophageal cancer and in patients with gastric cancer. ROC-analysis showed high sensitivity (91%) and specificity (85%) in the group with esophageal cancer but 68% and 85%, respectively, in patients with gastric cancer. Analysis of false negatives IgG-LysK in cancer patients showed that most patients had an advanced stage of cancer accompanied by metastases. Total IgG in the plasma of patients with false-negative IgG-LysK values was 30% lower than in samples with positive values, while the level of a2-antiplasmin was increased and the prothrombin time was shorter. These changes in blood homeostasis may be the reason for an increase in the proportion of false-negative values of the IgG-LysK coefficient. Circulatory IgG-LysK levels increase in the early stages of such cancers as gastric and esophageal cancers. Thus, when used in a panel with other more specific markers for these pathologies, this indicator can significantly increase the early detection of cancer.

## 1. Introduction

Gastric cancer, a prevalent malignant human tumor, ranks fifth in morbidity and third in mortality among all malignant neoplasms [1,2,3,4]. Similarly, esophageal cancer ranks eighth in prevalence and seventh in mortality [5,6,7,8,9]. Gastroscopy and biopsy are the “gold standard” for early diagnosis of these pathologies [10]. However, their use for screening tests may not always be possible. Tumor markers such as CA 19-9, CA 72-4, CEA [11], squamous cell carcinoma antigen (SCC), and tumor marker 2 (TM2) are commonly used to diagnose gastric and esophageal cancers, but they are not specific or sensitive enough [12,13,14].

Invasion and metastasis, the hallmarks of cancer aggressiveness, are accompanied by the activation of proteolytic enzymes. The degree of tumor aggressiveness is known to directly correlate with the activity of proteolytic enzymes in the tumor microenvironment. In addition to metalloproteinases, serine proteases, specifically plasmin (PM), play an active role in proteolytic processes within the tumor microenvironment [15]. PM is an activation component of the plasminogen (PLG) proteolytic cascade [16,17,18]. The PM, which belongs to the serine protease family, has great significant because it is capable of cleaving nearly any component present in the extracellular matrix or basal lamina [19]. Local invasion, metastasis, and angiogenesis are facilitated by the overexpression of the components of the PLG activation system in malignant tumors [19,20]. A tissue- or urokinase-type plasminogen activator (tPA or uPA) is the main factor responsible for generating PM from inactive PLG [21]. PM is involved in tumor metastasis by digesting tumor adjacent tissues, that creates space and nutrients for further malignant growth, invasion and metastasis [15].

Blood supply is crucial for the growth, promotion, and metastatic spread of cancer cells. New blood vessels are required to deliver oxygen and nutrients to the growing tumor mass and to remove cellular waste products. Thus, the tumor tissue contains a large number of capillaries and is abundantly supplied with blood [22].

IgG is one of the major blood plasma proteins. A significant increase in IgG has been detected in the microenvironment of lung cancer, esophageal cancer, and colon cancer tissues [23]. IgG being in the area of increased PM activity is expected to cause an increase in its proteolysis in the tumor microenvironment. It was reported that the PM splits IgG, forming an IgG variant with a free C-terminal lysin [24] (Figure 1).

PLG is a single-chain glycoprotein. It contains the *N*-terminal peptide, five homologous kringle domains, and the protease domain. After cleavage of the Arg561–Val562 specific bond by plasminogen activators, single chain PLG is transformed into the two-chain PN. The heavy chain (60 kDa) and the light chain (25 kDa) of PN are connected by two disulfide bonds. A specific feature of heavy chain kringle domains is the presence of lysine-binding sites, which provides binding to fibrin, α2-antiplasmin, cell receptors, and extracellular ligands, affinity to which differs among the kringles [25].

The PLG heavy chain (PLG-H) has Lys binding sites, which may bind to a free C-terminal Lys of IgG [26].

Earlier, we showed that the number of IgG isolated from plasma of healthy donors and connected to a PLG heavy chain significantly increases after plasmin treatment. After the removal of C-terminal lysine in IgG treated with carboxypeptidase, the amount of IgG bound to the heavy chain was significantly lower. Moreover, when lysine is added, the IgG level of the heavy chain is also significantly reduced [26,27]. Other works also show that heavy chains of plasminogen contain lysine-binding sites, allowing it to bind to protein fragments with free C-terminal lysine formed after proteolysis by plasmin [23,24,28,29].

We have previously investigated the level of G immunoglobulin degradation products in the blood serum for prostate and lung cancer. The evaluation of IgG fragments with the exposed C-terminal lysine was based on the identification of IgG interaction with lysine-binding cringles of the heavy plasmin chain in ELISA. In this assay, a heavy chain of plasmin adsorbed on 96-well polystyrene plates was used as a detector for IgG fragment with exposed C-terminal lysine (Figure 2).

Light blue kringles 1–5—lysine-binding sites; dark blue fragments—proteolytic IgG fragments with exposed lysine; red IgG—a conjugate of mouse monoclonal antibodies to human IgG with horseradish peroxidase; yellow ovals—horseradish peroxidase linked detection antibody.

The level of proteolytic fragments of IgG was significantly higher in patients with prostate cancer compared to healthy donors. In addition, patients with benign hyperplasia had levels of IgG fragments significantly lower than patients with prostate cancer. At the same time, a specific marker of prostate cancer PSA does not differentiate malignant tumor from benign [22,26]. The level of IgG proteolytic fragments in adenocarcinoma and squamous cell lung cancer was also significantly higher than in healthy donors [25,27].

Given that gastric and esophageal tumors are also solid malignant tumors and may exhibit increased proteolytic activity, the purpose of this study was to investigate the presence of immunoglobulin fragments with a free C-terminal lysine in the blood serum of patients with various types of these diseases.

## 2. Materials and Methods

The study included 43 patients with esophageal cancer and 68 patients with gastric cancer (Table 1). All of them underwent examination and treatment at the Federal State Budgetary Institution “N.N. Blokhin National Medical Research Center of Oncology” of the Ministry of Health of the Russian Federation. All procedures performed on patients and healthy donors comply with the ethical standards of the institution’s ethics committee and the Declaration of Helsinki (1964) and its subsequent amendments or comparable Ethical standards. Informed consent was obtained from all study participants. The clinical diagnosis in all patients was confirmed through morphological examination of the tumor based on the International Histological Classification (WHO, 2019) [25,26,30,31]. The control group consisted of 20 healthy donors, with 12 men and 8 women (60% and 40%, respectively), who were also examined at the N.N. Blokhin National Medical Research Center of Oncology. Before treatment, we collected blood serum and plasma samples using standard methods and stored them for 2 months at −80 °C.

***Preparation of heavy chain of plasminogen (Pg-H)*.** Plasminogen (PG) was obtained from a pool of human plasma as described [25,27]. Shortly after the plasma defrosted, 300 mL of aprotinin (Gordox, Gedeon Richter, Budapest, Hungary) (6000 IU) was added. Using affinity chromatography on Lys-Sepharose 4B (GE Healthcare Life Sciences, Uppsala, Sweden) at 4 °C, PG was separated from the plasma. Ammonium sulfate was used to precipitate fractions of protein peaks of PG that were eluted from the column using 0.2 μ 6-AHA (6-aminohexanoic acid, Acros Organics, Geel, Belgium), dialyzed, and lyophilized, and 12% PAAG SDS-electrophoresis was used to verify the preparations’ purity. Pg-H was produced by first activating PG and then reducing S-S bonds in an environment that prevented autolysis. In 0.1 M phosphate buffer (pH 8.0), 0.46 M iodoacetic acid was used to inhibit the resultant SH-groups after PG’s disulfide bonds were weakened by 0.25 mM mercaptoethanol for 20 min. Using affinity chromatography on Lys-Sepharose 4Β equilibrated with 0.1 M phosphate buffer (pH 8.0) containing 20 IU/mL of aprotinin, the light and heavy chains of PG were separated. After equilibrating the buffer, light chain PG (Pg-L) that had not bound to the carrier was eluted, and Pg-H that had adsorbed on the carrier was then eluted using a 0.2 M 6-AHA solution in the same buffer. Pg-H was purified by dialysis against water using a dialysis tube (12,000 Da). Pg-H was then lyophilized, and 12% PAAG SDS-electrophoresis was used to verify the preparations’ purity.

***ELISA of IgG fragments***. Pg-H adsorption was performed on a 96-well plate surface (Maxi Binding, (Seoul, Republic of Korea). Pg-H (5 μg/mL) in 0.1 Μ Na_2_Ρ\3/NaHΡ\3 buffer (pH 9.6) was introduced to each of the plate’s 100 μL wells, and the plates were incubated at 4 °C for the whole night. Following the solution’s removal, 200 μL of a 1% BSA (P Medicals, New Zealand) solution in PBS (137 mM NaCl, 2.7 mM KCl, 10 mM Na_2_HPO_4_, 1.8 mM KH_2_PO_4_; pѝ 7.4) was added to each well to block any remaining active sites, and the combination was then incubated for 12 h at room temperature. Following the blocking solution’s removal, the plates were kept at 4 °C and dried for 12 h at room temperature. Every experiment was run three times. The ratio of serum samples to PBS containing 0.5% BSA was 1:300. After adding 100 μL of each diluted sample to each of two plate wells containing immobilized Pg-H, the wells were incubated for one hour at 37 °C. Following the solution removal, the wells were cleaned four times using 200 μL of PBS with 0.05% Tween-20 (PBS-T). Next, 100 μL of the mouse monoclonal antibody to human IgG conjugate working solution was added, and the combination was incubated at 37 °C for one hour. After that, the wells were cleaned four times using 200 μL of PBS-T, and 100 μL of a tetramethylbenzidine and hydrogen peroxide solution was added to each well. The wells were then incubated for 15 minutes at 37 °C in the dark. A 2 μ H2SO4 solution was added to stop the reaction, and a multichannel spectrophotometer (Bio-Rad, Hercules, CA, USA) was used to measure the solution’s optical absorbance at 450 nm (A450). The IgG-fragment level (IgG-LysK) was expressed using the ratio of the optical density of the patient sample obtained in the ELISA to the cutoff. The cutoff was calculated by ROC analysis of the optical density of patient samples versus the optical density of healthy donors. IgG-LysK values less than 1 were considered negative, and values greater than 1 were positive values. The total content of IgG in the serum was determined by ELISA kit (#EH0417, Wuhan Fine Biotech Co., Ltd., Wuhan, China). Samples of blood serum were diluted to 1/1000 for testing.

***Analysis of hemostasis.*** Plasma samples were characterized on an Sysmex CA-1500 automatic analyzer (SYSMEX CORPORATION, Kobe, Japan) according to prothrombin time (PT) (“Thromborel S”, Siemens (10 × 10 mL) OUHP49) and fibrinogen concentration (Fibrinogen) (“Multifibren U”, Siemens (10 × 5 mL) 10446691). Plasma samples were characterized by the optical method according to the main indicator of the plasminogen/plasmin system: plasminogen activity (“Reachrom-plasminogen”, #FA-2, SPD Renam, Moscow, Russia), α2-antiplasmin activity (“Reachrom-α2-antiplasmin”, #FA-3, SPD Renam, Moscow, Russia) and protein C activity (“Reachrom-protein C”, #FA-5, SPD Renam, Moscow, Russia)

Statistical analysis was performed using the Statistica (Ver.12, StatSoft Inc., Tulsa, OK, USA). The distribution of values of the studied indicators differed significantly from the Gaussian distribution, so the median and quartiles were calculated. To compare indicators and analyze their relationships, we performed non-parametric Mann–Whitney tests. An analysis of the informational value of the level of proteolytic IgG fragments in the blood serum of patients was conducted by constructing the curves of the ROC and calculating the area below them (AUC). Differences and correlations were considered statistically significant at *p* < 0.05.

## 3. Results

According to histological and clinical–morphological analysis, the majority of patients (88%) with esophageal cancer were diagnosed with squamous cell carcinoma. However, in the group of patients with gastric cancer, the majority of patients (84%) were diagnosed with adenocarcinoma (see Table 1).

The study revealed that the serum of healthy donors in the control group had a significantly lower median content of IgG-LysK compared to the esophageal cancer group (*p* < 0.0001) and the gastric cancer group (*p* = 0.003) (see Table 2). However, false-positive values of IgG-LysK were detected in 15% of the control group. False negatives were observed in 12% of patients with esophageal cancer and 32% of patients with gastric cancer.

We utilized ROC-analysis to assess the information value of the level of proteolytic IgG fragments in blood serum. It showed 91% sensitivity and 85% specificity for patients with esophageal cancer and 68% and 85% for gastric cancer (Figure 3A,B).

We identified a false-negative IgG-LysK coefficient in 22 patients with gastric cancer; 20 samples of them had stages III and IV and distant metastases.

Next, we attempted to investigate the reason for the increase in IgG-LysK false-negative values in patients with gastric cancer. We hypothesized that this could be linked to a reduction in the overall number of immunoglobulins in advanced stages of the disease and immunodeficiency due to cancer progression related.

Our analysis showed that in patients with stages III–IV of the disease, an average of total IgG level with a false-negative IgG-LysK coefficient was 29% lower compared to that in patients with a positive coefficient (see Table 3, Figure 4).

Since plasminogen takes an active part in blood coagulation, we compared the concentrations of fibrinogen in the blood of patients with gastric cancer related to both positive and false-negative IgG-LysK coefficients. Although the concentration of fibrinogen increased in the blood of patients with false-negative IgG-LysK coefficients, we did not reveal significant differences.

Next, we compared prothrombin time in these groups and detected that it reduced more significantly in the group with false-negative IgG-LysK coefficients that in the ones with positive coefficients (see Table 4, Figure 5).

Since plasmin, a derivative of plasminogen, is involved in the formation of fragments of immunoglobulins with C-terminal lysine, the activity of plasmin inhibitor, α2-antiplasmin, was assessed in samples of gastric cancer patients with positive and false-negative IgG-LysK coefficients. The data obtained indicated that the α2-antiplasmin activity in samples with false-negative IgG-LysK coefficients was by 27% higher than in the positive ones (see Table 5, Figure 6).

There are no differences in the activity of protein C in the positive and false negative samples.

## 4. Discussion

Clinical manifestations of gastric tumors are scarce and most often observed in advanced cases [30]. WHO statistics demonstrate that early-stage gastric cancer is detected in only 10% of cases. As opposed to this, about 75% of initially diagnosed gastric cancer are revealed in stages III–IV, on which more than 80% of patients with initial examination of the tumor having metastases to regional lymph nodes [25,26,29,30,31,32,33,34,35].

Early diagnosis therefore remains a very important problem at present. The process of initiating malignant lesions is associated with increased activity of tumor proteases, which promotes its growth in the surrounding tissues. With the help of proteases, the tumor cleaves various components of the surrounding tissues, resulting in the emergence of proteolytic products that can get into circulation and indicate the appearance of carcinogenesis. However, in the early stages of tumor development, the concentration of known specific markers in the blood serum is insufficient for sensitivity of existing diagnostic tests. Despite the understanding that malignancy is associated with the activation of proteases and the appearance of proteolytic products, there are no universal diagnostic markers. Among many proteases that support tumor survival, plasminogen is one of the most common proteolytic enzymes in the body. Its active form is plasmin, which can cleave the human IgG [25,26]. We have suggested that a tumor damaging the surrounding tissue is capable of activating plasmin that dissects IgG. Proteolytic fragments of IgG with available lysine at the C-terminal may circulate and be detected by ELISA. To normalize ELISA values, we introduced an IgG-LysK coefficient reflecting the level of proteolytic fragments IgG in the patients’ serum.

The results of our research showed that the level of proteolytic fragments of IgG in blood serum of patients with esophageal cancer (*p* < 0.0001) and gastric cancer (*p* = 0.003) is significantly higher than that in healthy volunteers (Table 2). According to the ROC analysis, the sensitivity (SN) of the classification of patients with esophageal cancer and in healthy controls (*n* = 43 and *n* = 20 respectively) was 91% sensitivity (SN), 85% specificity (SP), and an area under the curve (AUC) of 0.903 (Figure 3A). At the same time, for patients with gastric cancer and in healthy controls, SN was 68% at 85% SP and AUC of 0.719.

In analyzing the reasons for the decreased sensitivity of the test in the diagnosis of gastric cancer, it was found that the false negative IgG-LysK coefficient was observed mainly in patients with late stages (III–IV), which accounted for 60% of the total number of observations in the group. Out of 22 false-negative IgG-LysK samples, 20 patients with gastric cancer (29% of all patients with gastric cancer) had distant metastases (Table 1). The same pattern was observed in the esophageal cancer group. Patients with advanced metastases (12%) showed low levels of IgG fragments in the blood serum and false negative IgG-LysK values.

Gastric cancer is known to rank second after pancreatic cancer in the incidence of venous thromboembolic complications [28,29,32,33]. Blood plasma samples of patients with false negative IgG-LysK values had a reduced PT time (*p* < 0.001, Table 4). This may indicate partial blockage of the anticoagulation system and a tendency to thrombosis in patients with false IgG-LysK values.

According to the literature, patients with stage III–IV cancer are more likely to experience thrombosis than patients with stage I–II cancer. Moreover, some patients in late stages show a decrease in plasmin activity and activation of plasminogen inhibitors α2-antiplasmin and plasminogen activator inhibitor type 1 [PAI-1] [30,31,32,33,34,35,36,37]. In our research, we also showed that patients with false-negative IgG-LysK values had significantly higher levels of α2-antiplasmin than patients with positive IgG-LysK. Thus, as cancer progresses, disruptions arise within the hemostatic system, and patients are more likely to develop thromboembolic complications.

Additionally, total IgG levels in serum of patients with late stages and distant metastases and with the false-negative IgG-LysK coefficient were significantly lower than in those patients with high values of IgG-LysK. A lot of samples with false-negative IgG-LysK values in patients with gastric cancer can be explained by a decrease in both the plasmin activity and the level of immunoglobulins. The last in turn leads to a decrease in immunoglobulin fragments with C-terminal lysine.

The high sensitivity of the test in patients with esophageal cancer can be explained by the fact that the number of late-stage patients with distant metastases was significantly lower (only 12% in the group) than in patients with advanced gastric cancer (29%). It should be noted that samples of patients diagnosed with T3-4 and regional metastases to lymph nodes (N) also gave positive IgG-LysK values. False-negative values were given only with remote metastases (M). Thus, significant changes in hemostasis are likely to occur after the emergence of remote metastases. So, IgG-LysK sensitivity and specificity vary significantly between early and late stages of cancer due to distant metastases in late stages. In order to arrive at a distinct conclusion in the future, it is necessary to conduct additional research with a larger group of patients.

The high IgG-LysK values in patients with early nonmetastasized cancers identified in this study are consistent with data from prostate cancer [26] and lung cancer [27]. Thus, if diagnosed, the decline may indicate the beginning of metastasis. In addition, according to the literature, IgG contains C-terminal lysine on its heavy chain [38]. However, healthy volunteers have a low IgG bound to a heavy plasminogen chain, which can be explained by the inaccessibility of C-terminal lysine. After plasmin treatment, IgG interaction with heavy chains of plasminogen increases dramatically [26]. It is possible that the conformation of the Fc fragment of IgG changes during proteolysis, resulting in the exposure of C-terminal lysine.

The presence of defective antibodies in circulation can be the reason for the lack of effect when using antibodies during immunotherapy as well as for a decrease in sensitivity of a number of oncomarkers. It is also important to emphasize the relationship between plasminogen activation in tumors and the presence of IgG fragments whose effects on tumor growth have not been studied. Thus, IgG fragments can not only be an indicator of proteolysis activation in tumors but also a possible factor influencing tumor growth.

It is possible that the dynamics of the level of IgG fragments in circulation depend not only on the stage of cancer but also on the type of tumor and its localization. Elevated levels of IgG-LysK in circulation have been found in esophageal, gastric, prostate, and lung cancers. IgG-LysK has not been tested in other types of tumors.

In the last few years, diagnostic and predictive techniques have taken a leap with the use of artificial intelligence [39]. The inclusion of our results in the database used by artificial intelligence may increase the reliability of diagnostics of malignant neoplasms. Assessment of the level of proteolytic IgG fragments in the blood serum in addition to existing specific tumor markers can increase the sensitivity and specificity of diagnosis in the early stages, differentiate a malignant tumor from a benign, one and choose a management strategy [40]. This is especially important in early cancer stages without clinical manifestation.

Recent advances in the molecular characterization of gastric cancer, including diverse histological and genomic subtypes, have led to the development of targeted therapy and immunotherapy that improves patient prognosis. Molecular profiling conducted for the development of new biomarkers for effective stratification of patients and personalized treatment. Biomarkers like CEA, CA199, CA125, Claudin 18.2, FGFR2, Her2, p53, MSI, RAS, etc., play an important role in early detection and treatment of upper-gastrointestinal cancers, helping to define treatment strategies and predict clinical outcomes. Nevertheless, reliable circulating serum markers are preferable to markers derived from tumor tissue bioptates given the invasive biopsy procedure in patients and the associated difficulty of repeated histological studies [41].

Biomarkers that predict esophageal and gastric adenocarcinoma, like HER2, dMMR/MSI-H, and PD-L1, are used to guide treatment decisions for patients. It is crucial to continuously evaluate new and promising biomarkers and incorporate them into clinical practices to optimize treatment choices and enhance patient outcomes [42]. A more detailed comparison of IgG-LysK with existing markers for esophageal and gastric cancers to establish its relative advantage or complementarity could be beneficial for future research in finding promising biomarkers and panels of markers.

With the results of our research, it is possible that the IgG-LysK indicator will emerge as a new serum biomarker that will enhance the diagnostic accuracy and treatment outcomes of patients. It is necessary to conduct additional research focused on the correlation of immunoglobulin fragments with a free C-terminal lysine in the blood serum of patients with different stages and subtypes of the diseases to confirm its clinical use and determine its role in clinical practice.

**Limitations of research.** Early detection of cancer in most cases is a random discovery. It is therefore difficult to collect a representative sample. In our case, the number of observations of early-stage cancer without metastases is not sufficient to draw definitive conclusions about the utility of IgG-LysK in early diagnosis. False-negative results, especially in patients with advanced gastric cancer and distant metastases, raise concerns about IgG-LysK reliability as a standalone marker. In addition, the level of IgG-LysK in diseases that make a differential diagnosis with oncological diseases difficult is not investigated. Therefore, in order to confirm the results of this work, it is necessary to carry out a study of the level of IgG-LysK in a larger cohort with early stages of the disease and non-oncological diseases.

## 5. Conclusions

IgG-LysK levels in blood serum are significantly affected by the state of the fibrinolytic system and the total IgG level, which is the reason for the low IgG-LysK values in some patients with distant metastases in the late stages of gastric or esophageal cancer.

Circulatory IgG-LysK levels increase in the early stages of such cancers as gastric and esophagus. Thus, when used in a panel with other more specific markers for these pathologies, this indicator can significantly increase the early detection of cancer.

## Figures and Tables

**Figure 1 diagnostics-14-01396-f001:**
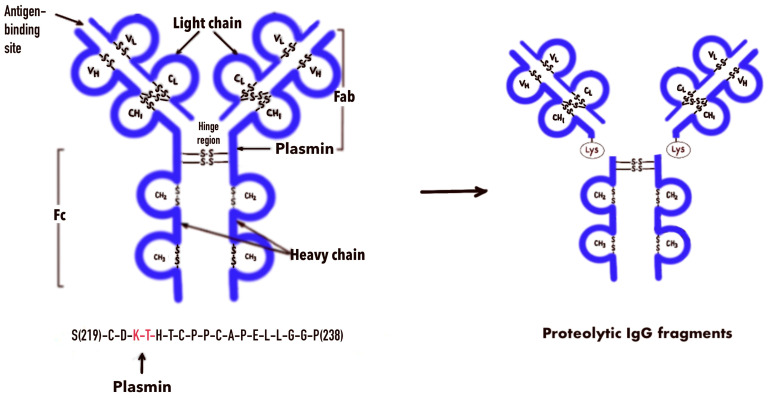
Scheme of proteolysis of IgG by plasmin.

**Figure 2 diagnostics-14-01396-f002:**
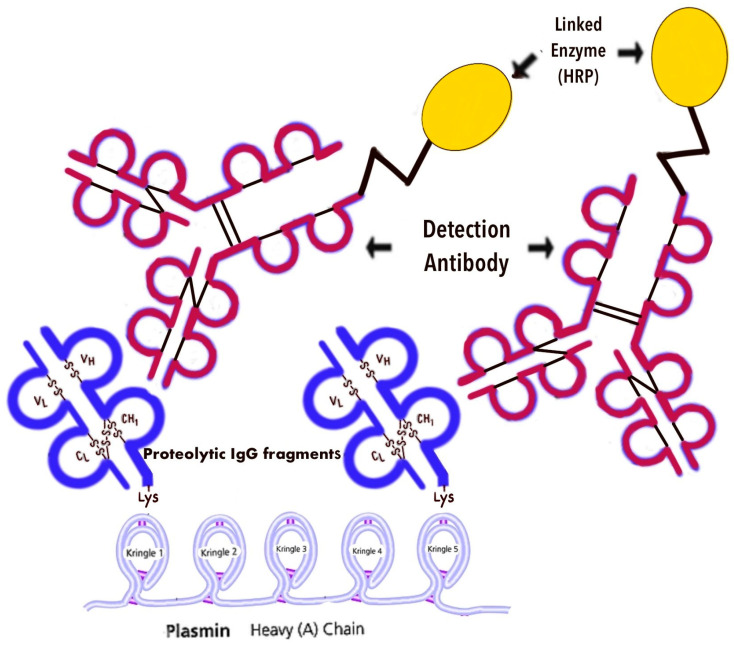
The evaluation of IgG fragments with the exposed C-terminal Lysine by ELISA.

**Figure 3 diagnostics-14-01396-f003:**
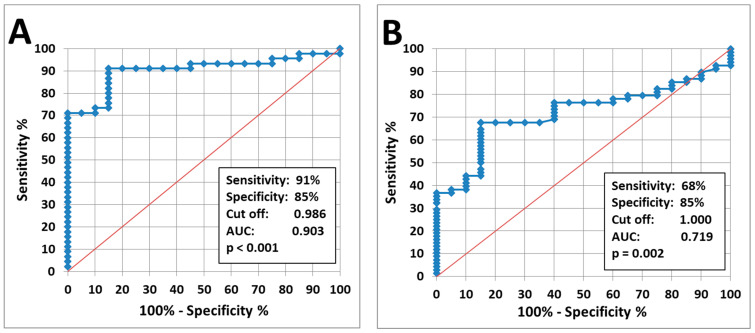
ROC analysis of IgG-LysK coefficient: (**A**) patients with esophageal cancer vs. healthy donors, (**B**) patients with gastric cancer vs. healthy donors.

**Figure 4 diagnostics-14-01396-f004:**
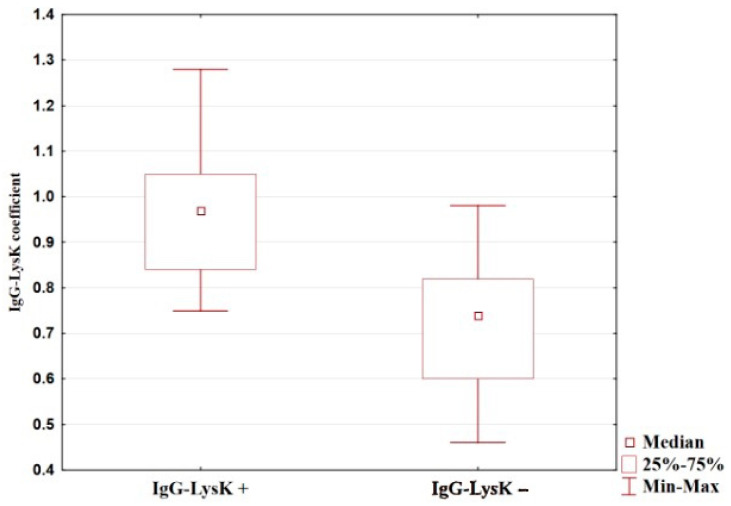
The level of IgG in the blood of patients with gastric cancer with positive and false-negative IgG-LysK coefficients.

**Figure 5 diagnostics-14-01396-f005:**
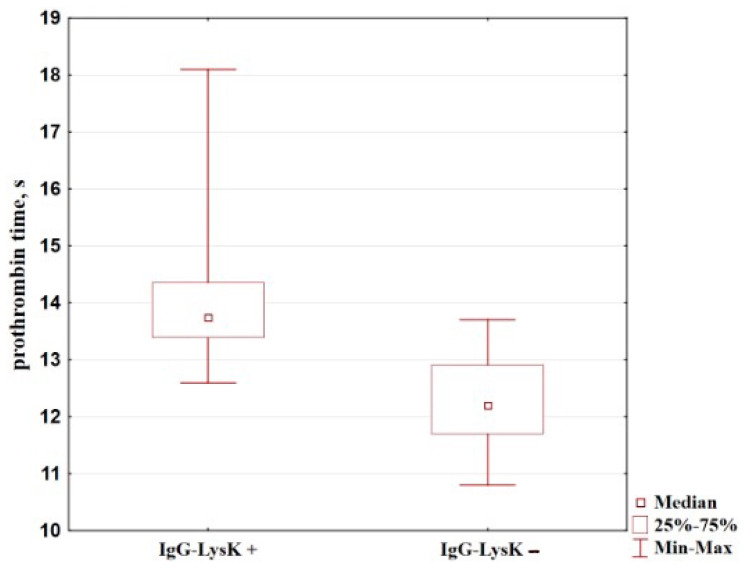
Comparison of prothrombin time in patients with gastric cancer with positive and false-negative IgG-LysK coefficients.

**Figure 6 diagnostics-14-01396-f006:**
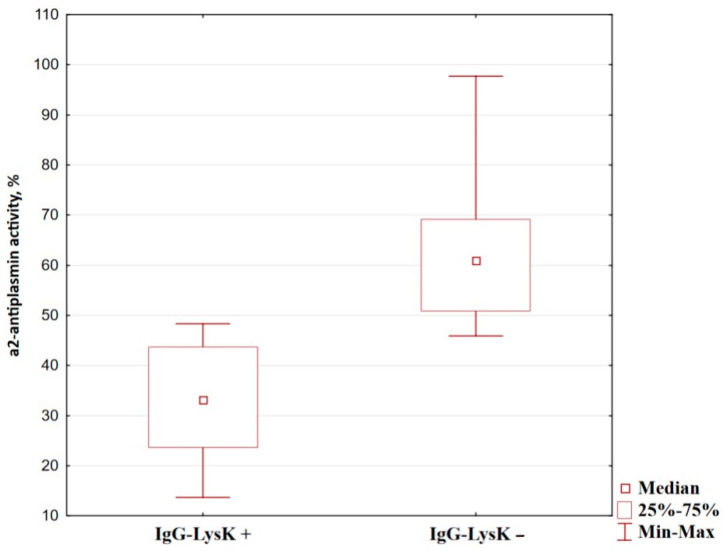
The α2-antiplasmin activity (%) in the blood plasma of patients with gastric cancer with positive and false-negative IgG-LysK coefficients.

**Table 1 diagnostics-14-01396-t001:** Characteristics of selected populations.

Cohort	Gastric Cancer		Esophageal Cancer		Control Group	
	Male	Femal	Male	Female	Male	Female
*n*	36	32	31	12	12	8
Age, years old, min–max (median)	20–80 (62)	34–83 (66)	39–82 (63)	33–70 (63)	48–71 (62)	26–70 (62)
Adenocarcinoma	*n* = 31	*n* = 26	*n* = 4	*n* = 1	-	-
Signet ring cancer	*n* = 5	*n* = 6	-	-	-	-
Squamous cell carcinoma	-	-	*n* = 27	*n* = 11	-	-
Stage I–II	*n* = 15	*n* = 12	*n* = 13	*n* = 6	-	-
Stage III–IV	*n* = 21	*n* = 20	*n* = 18	*n* = 6	-	-
Tumor size T1-2	*n* = 7	*n* = 8	*n* = 8	*n* = 3	-	-
Tumor size T3-4	*n* = 29	*n* = 24	*n* = 23	*n* = 9	-	-
Lymph nodal spreadN0	*n* = 13	*n* = 13	*n* = 11	*n* = 6	-	-
Lymph nodal spread *n*+	*n* = 23	*n* = 19	*n* = 20	*n* = 6	-	-
Metastasis M+	*n* = 9	*n* = 11	*n* = 4	*n* = 1	-	-

**Table 2 diagnostics-14-01396-t002:** Statistical analysis of the content of IgG-LysK in the serum of the control group as well as patients with the esophageal and gastric cancers.

Group	Number of Cases	IgG-LysK		
		Median; Quartiles Me (Q1:Q3)	*p*	
**1. Control group**	20	0.86 (0.74; 1.00)	1–2	1–3
**2. Esophageal** **cancer**	43	1.71 (1.09; 4.03)	<0.0001 *	
**3. Gastric cancer**	68	1.08 (0.90; 1.60)		0.003 *

(*) *p* < 0.05.

**Table 3 diagnostics-14-01396-t003:** The level of IgG (mg/mL) in the blood of patients with gastric cancer with positive and false-negative IgG-LysK coefficients.

Group	Number of Cases	Median; Quartiles Me (Q1:Q3)	*p*	
**IgG-LysK−**	11	7.6 (5.6: 8.8)	IgG-LysK−/IgG-LysK+	=0.001 *
**IgG-LysK+**	11	11.1 (9.5: 12.3)		

(*) *p* < 0.05.

**Table 4 diagnostics-14-01396-t004:** Comparison of prothrombin time (s) in patients with gastric cancer with positive and false-negative IgG-LysK coefficients.

Group	Number of Cases	Median; Quartiles Me (Q1:Q3)	*p*	
**IgG-LysK** **−**	20	12.20 (11.7:12.9)	IgG-LysK−/IgG-LysK+	<0.0001 *
**IgG-LysK+**	44	13.75 (13.4:14.3)		

(*) *p* < 0.05.

**Table 5 diagnostics-14-01396-t005:** The α2-antiplasmin activity (%) in the blood plasma of patients with gastric cancer with positive and false-negative IgG-LysK coefficients.

Group	*n*	Median; Quartiles Me (Q1:Q3)	*p*	
**IgG-LysK−**	15	67.3 (53.8:78.8)	IgG-LysK−/IgG-LysK+	<0.0001
**IgG-LysK+**	14	33.2 (23.9:43.2)		

## Data Availability

The data presented in this study are available upon request from the corresponding author.

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
