# Peer review of "Circulating IgG Fragments for Gastric Cancer and Esophageal Cancer"

_diagnostics, 2024, doi:10.3390/diagnostics14131396_

Round 1

Reviewer 1 Report (Previous Reviewer 1)

Comments and Suggestions for Authors

 This is a very interesting topic to study IgG-LysK as a new biomarker of gastric and esophageal cancers.  The manuscript can be improved on several points before publication.

1. Gastric cancer is a heterogeneous disease with diverse histological and genomic subtypes.  The study should focus on the most common gastric and esophageal tumors of the pathological type of adenocarcinoma with potential increased proteolytic activity, the study focused on the correlation of immunoglobulin fragments with a free C-terminal lysine in the blood serum of patients with different stages of the diseases, the squamous carcinoma might have different pathogenesis of the cancer biology.

2. The authors mentioned but not list the correlation of the panel with other more specific markers (CEA, CA199, CA125, Claudin 18.2, FGFR2, Her2, p53, MSI, RAS etc) for these pathologies as early detection of the upper GI cancers.

3.  The author may discuss the growing number of clinical trials based on novel biomarkers have demonstrated the efficacy of targeted therapies alone or in combination with conventional chemotherapy. Enrichment design clinical trials of targeted therapies against FGFR2b and claudin 18.2 have demonstrated efficacy in unresectable advanced gastric cancer. Nonetheless, it is essential to continuously validate promising molecular biomarkers and introduce them into clinical practice to optimize treatment selection and improve patient outcomes.

Author Response

Comment 1. Gastric cancer is a heterogeneous disease with diverse histological and genomic subtypes.  The study should focus on the most common gastric and esophageal tumors of the pathological type of adenocarcinoma with potential increased proteolytic activity, the study focused on the correlation of immunoglobulin fragments with a free C-terminal lysine in the blood serum of patients with different stages of the diseases, the squamous carcinoma might have different pathogenesis of the cancer biology.

Response 1: [Type your response here and mark your revisions in red] Thank you for pointing this out. We agree with this comment. Therefore, we have added some sentences for discussion of advances in the molecular characterization of gastric cancer [lines 341-343] and the necessity of the study focused on the correlation of immunoglobulin fragments with a free C-terminal lysine in the blood serum of patients with different stages of the diseases [lines 353-356].

Thank you for your detailed consideration of our article. In the analysis of samples from patients with stomach cancer, 81% were patients with adenocarcinoma and 19% with signet ring cancer (57 and 11), while those with esophageal cancer were inversely 87% with squamous cell cancer and 13% with adenocarcinoma (38 and 5). In order to compare the proteolytic activity of adenocarcinoma and squamous-cell gastric cancer, as well as esophagus, further studies are needed with a large number of observations in each group.

Comment 2.  The authors mentioned but not list the correlation of the panel with other more specific markers (CEA, CA199, CA125, Claudin 18.2, FGFR2, Her2, p53, MSI, RAS etc) for these pathologies as early detection of the upper GI cancers.

Response 2: We agree with this comment. We have pointed to an important role of these biomarkers in early detection and treatment of the upper gastrointestinal cancers [lines 345-347]. Future research may aim to improve the diagnostic accuracy of the panel of markers by examining the correlation between levels of IgG-LysK and previously known markers for gastric cancer.

Comment 3.  The author may discuss the growing number of clinical trials based on novel biomarkers have demonstrated the efficacy of targeted therapies alone or in combination with conventional chemotherapy. Enrichment design clinical trials of targeted therapies against FGFR2b and claudin 18.2 have demonstrated efficacy in unresectable advanced gastric cancer. Nonetheless, it is essential to continuously validate promising molecular biomarkers and introduce them into clinical practice to optimize treatment selection and improve patient outcomes.

Response 3: Thank you for pointing this out. We have discussed this theme in lines 347-356. In addition, the samples we collected were from primary patients. The drop in our test scores in patients with remote metastases was unexpected. Further study of changes in hemostasis in such patients, in addition to the data we have received, may be useful for monitoring and predicting the further course of these diseases.

We are really greatful for your notices that help to improve our paper. 

Reviewer 2 Report (Previous Reviewer 2)

Comments and Suggestions for Authors

The authors have responded to my comments. The answers are only explanatory and have not changed the manuscript. I am satisfied that the authors are contemplating studying patients with gastritis and esophagitis. The shrunken sample size has been explained adequately. 

Author Response

Comment 1. The authors have responded to my comments. The answers are only explanatory and have not changed the manuscript. I am satisfied that the authors are contemplating studying patients with gastritis and esophagitis. The shrunken sample size has been explained adequately.  Response 1: Thank you very much for your appreciated comments. Your notices have allowed us to make significant revisions to our article and improve its content.

This manuscript is a resubmission of an earlier submission. The following is a list of the peer review reports and author responses from that submission.

Round 1

Reviewer 1 Report

Comments and Suggestions for Authors

This is a well-designed study that expands the results based on the previous investigation to quantify the level of G immunoglobulin degradation products in the blood serum for prostate and lung cancer.

The evaluation of IgG fragments with the exposed C-terminal lysine was based on the identification of IgG interaction with lysine-binding cringles of the heavy plasmin chain in ELISA.  The authors concluded the emergence of defective antibodies in the early stages of stomach and esophageal cancers as well as in the prostate, and lung cancers. 

Author Response

Thank you very much for the careful analysis of our material. It is very important for us to get feedback on our research and possibly stimulate the scientific community’s interest in IgG fragments not only as an indicator of proteolysis activation in tumors, but as a possible factor influencing tumor growth. In the future we plan to collect a representative number of samples at different stages of cancer, as well as samples from patients with benign diseases posing difficulties in differential diagnosis. We hope that our results may be of interest to the global scientific community, as the appearance of IgG fragments at the earliest stages of malignacy can signal the onset of the cancer process.

Reviewer 2 Report

Comments and Suggestions for Authors

Authors have estimated IgG degradation products (IgG-LysK) in 43 patients with squamous cell carcinoma esophagus (EC) and 68 patients with adenocarcinoma of the stomach (GC) and compared it with 20 healthy controls. IgG fragments were higher in EC and GC patients when compared with controls. For EC, sensitivity and specificity were high (91% and 85%), and for GC 68% and 85%. The authors point out that these results may be used as a screening test for diagnosis of EC in the community as other markers are not specific and diagnosis needs invasive testing like endoscopy. 

I have the following comments to make:

1. IgG fragments from tumors are not specific for EC and GC and can be high in any tumors where such processing can occur like lung or prostrate etc. 

2. The authors did not include patients with esophagitis and gastritis in the control group. Will it occur in such benign diseases is not known. 

3. The number of patients and controls is too small to make any suggestions on the value of the screening test. 

Author Response

Thank you very much for taking the time to review this manuscript. Please find the detailed responses below.

Comments 1: IgG fragments from tumors are not specific for EC and GC and can be high in any tumors where such processing can occur like lung or prostrate etc.

Response 1: Thank you for pointing this out. We agree with this comment. Thank you very much for the in-depth analysis of our material. Malignant tumors are characterized by high activity of serine proteases, in particular plasmin, and metalloprotases. The process of beginning malignation is associated with an increase in the protease activity of the tumor, which contributes to its growth into the surrounding tissue. With the help of proteases, the tumor cleaves various components of the surrounding tissue, resulting in the emergence of proteolytic products that can get into circulation and indicate the onset of cancerogenesis. However, in the early stages of tumor development, the concentration in the blood of known specific serum markers is not sufficient for sensitivity of existing diagnostic tests. Despite the understanding that malignancy is related to the activation of proteases and the appearance of proteolytic products, universal diagnostic markers, which would indicate the beginning of the tumor process in the body and would increase the detection of cancer at the earliest stages is not currently proposed. Among the many proteases that support tumor survival, plasminogen is one of the most common proteolytic enzymes in the body. Its active form is plasmin, which can split human IgG. We assumed that the tumor damaging the surrounding tissue was able to activate plasmin, which cleaves IgG. Proteolytic fragments of IgG, exposed lysine at the C-terminal end, may enter the circulation. We have proposed an ELISA method to determine the level of such IgG fragments, due to their ability to bind to lysine-binding cringles of the heavy plasminogen chain. We have previously shown an increase in levels of serum IgG fragments in patients with prostate cancer and lung cancer. However, this work found that in stomach cancer, the number of false negative patients increases significantly in later stages with metastasis. It is possible that the dynamics of the IgG fragment level in circulation depends on the type of tumor, its localization and stage of cancer. The level of proteolytic fragments of IgG in serum of patients with other types of cancer has not been investigated. Despite the limited number of observations, we believe that our results may be of interest to the global scientific community, as the appearance of IgG fragments at the earliest stages of malignancy may signal the onset of the cancer process, when there are too few specific markers. Thus, defining this indicator together with specific oncomarkers can be useful for cancer diagnosis and monitoring relapse after therapy.

Comments 2: The authors did not include patients with esophagitis and gastritis in the control group. Will it occur in such benign diseases is not known.

Response 2: We absolutely agree with your comment about the need to investigate benign diseases to clarify the question of the possibility of differential diagnosis. In our previous work we showed a significant increase in the level of IgG fragments in the serum in patients with prostate cancer compared to benign hyperplasia and healthy donors. Given the difference in serum levels of IgG between cancer patients and the control group, we plan to examine serum samples from patients with gastritis, esophagitis and benign tumors. We hope that the scientific community will pay attention to the relationship between the activation of plasminogen in the tumor and the appearance of fragments IgG, the effect of which on the growth of the tumor has not been studied.

Comments 3: The number of patients and controls is too small to make any suggestions on the value of the screening test.

Response 3: We absolutely agree with you, our results are indeed limited to a small sample to suggest their value in screening tests. However, the early detection of cancer in most cases is an accidental finding. Therefore, there are some difficulties in collecting a representative sample. Since some of the results from advanced stages of stomach cancer and distant metastases were somewhat unexpected to us, we plan to collect a representative number of samples at different stages in the future. It is very important for us to get feedback on our research and possibly stimulate the scientific community’s interest in IgG fragments not only as an indicator of proteolysis activation in tumors, but as a possible factor influencing tumor growth. Thank you very much for your interest in our research and understanding of the limitations of our results. We will take your comments into account in our further work, it will help to improve our results.